# The role of phosphorus in the solid electrolyte interphase of argyrodite solid electrolytes

Matthew Burton [1,2], Ben Jagger [1], Yi Liang [1], Joshua S. Gibson [1], Jack Aspinall [1], Zhongdui Long [1,2], Jack E. N. Swallow[1], Robert S. Weatherup [1,3] & Mauro Pasta [1,2] ✉

The solid electrolyte interphase that forms on $Li_6PS_5Cl$ argyrodite solid electrolytes has been reported to continually grow through a diffusion-controlled process, yet this process is not fully understood. Here, we use a combination of electrochemical and X-ray photoelectron spectroscopy techniques to elucidate the role of phosphorus in this growth mechanism. We uncover how $Li_6PS_5Cl$ can decompose at potentials well above the full reduction to $Li_3P$, forming partially lithiated phosphorus species, $Li_xP$. We provide evidence of a gradient of $Li_xP$ species throughout the solid electrolyte interphase and propose a growth mechanism in which the rate-determining step is the diffusion of lithium through $Li_xP$. We predict continuous solid electrolyte interphase growth as long as metallic lithium is present and a $Li_xP$ percolation pathway exists, highlighting the importance of understanding and engineering solid electrolyte interphase composition and nanostructure in solid-state batteries. We believe that this growth mechanism would apply to any solid electrolyte interphase that can contain partially lithiated phosphorus, or potentially any lithium alloy.

The transition away from fossil fuels is driving an ever-growing demand for batteries[1]. One key area of demand is transportation, where increased energy density is crucial for improving vehicle range. Lithium metal negative electrode batteries hold significant promise in this regard, with the highest energy density achieved in a zero-lithium-excess (anodeless) configuration[2]. However, non-uniform lithium plating during charging compromises cell lifespan and safety when used with a liquid electrolyte[3]. In theory, this issue can be mitigated by applying a homogeneous stack pressure with a solid electrolyte (SE) with sufficient mechanical strength[4].

Several Li-ion conducting SEs have been discovered[5]. Oxides exhibit the lowest theoretical reduction potentials and, therefore, the greatest stability against Li metal negative electrodes[6]. However, their ionic conductivities have so far been too low to be used as the positive

electrode electrolyte in practical applications[5], whilst the oxide's poor solid-solid contacts between the Li metal and a rigid oxide solid electrolyte hinder its use as a separator[7]. In contrast, softer sulphides have demonstrated ionic conductivities exceeding those of liquid electrolytes but suffer from limited electrochemical stability windows and poor compatibility with metallic lithium[5].

Among sulphide solid electrolytes, argyrodites–composed of or closely related to $Li_6PS_5Cl$–are considered among the most promising due to their enhanced stability compared to other sulphides[8]. Despite the theoretical reduction potential of $Li_6PS_5Cl$ being 1.71 V vs. $Li_+/Li$[6], the solid electrolyte interphase (SEI) formed between $Li_6PS_5Cl$ and lithium was considered passivating, with a thickness on the order of nanometres[9]. Work by the Janek group challenged this assumption, reporting continuous SEI growth, which they described using the

[1]Department of Materials, University of Oxford, Oxford, United Kingdom. [2]The Faraday Institution, Quad One, Harwell Science and Innovation Campus, Didcot, United Kingdom. [3]Diamond Light Source, Harwell Science and Innovation Campus, Didcot, United Kingdom. ✉ e-mail: mauro.pasta@materials.ox.ac.uk

Wagner model for diffusion-controlled solid-state reactions[10–12]. Yet the underlying diffusion mechanism was not understood. They also estimated the SEI thickness to be on the order of hundreds of nanometres using Time-of-Flight Secondary Ion Mass Spectrometry (ToF-SIMS)[13], consistent with measurements from our group obtained via in situ X-ray photoelectron spectroscopy (XPS)[14].

The practical implications of this finding are critical for the viability of solid-state batteries, as continuous SEI growth leads to increased cell impedance[10], electrolyte and lithium consumption, and exacerbation of the chemo-mechanical degradation of the solid electrolyte, ultimately limiting battery life[15]. It is therefore imperative to gain a deeper understanding of the SEI and the mechanism that underpins its growth.

This work investigates the mechanism behind the continual growth of the Li-argyrodite SEI, linking its behaviour to SEI composition. The conductivity of the SEI is assessed through a combination of coulometric titration time analysis (CTTA) and three-electrode potentiostatic electrochemical impedance spectroscopy (PEIS). The evolution of the SEI's chemical composition is characterised using virtual electrode plating X-ray photoelectron spectroscopy (VEP-XPS), while non-destructive depth profiling of the SEI is performed via soft and hard X-ray photoelectron spectroscopy (SOXPES and HAXPES). Finally, the electrochemical stability of the SEI is examined using cyclic voltammetry (CV).

## Results

To investigate the origin of continuous SEI growth, we performed a modified version of the coulometric titration time analysis (CTTA) technique introduced by the Janek group (Fig. 1a)[11]. We incorporated an additional potentiostatic electrochemical impedance spectroscopy (PEIS) step between lithium plating cycles, using a three-electrode setup with an InLi-In ring reference electrode[16], specifically designed to minimise impedance artefacts (Fig. 1b, inset). Our results show a

similar rate of Li consumption (Fig. 1b)[11]. After ~ 900 h, 50 μAh cm$^{-2}$ of capacity is consumed, which would correspond to an SEI thickness of ≈450 nm (assuming a compact mixture of Li$_2$S, LiCl and Li$_3$P and the absence of gaseous products)[11]. The initial 15 h do not appear to follow a linear relationship between accumulated charge and the square root of time, suggesting this initial period may not be limited by diffusion. Thereafter, however, a strong linear fit is observed with an $R^2$ value of > 0.998.

Impedance measurements (sinus amplitude: 10 mV, frequency range: 400 kHz to 10 Hz) were taken once all the plated lithium had been consumed-i.e., when the 50 mV vs. Li$^+$/Li cutoff was reached-but before the next lithium plating step (Fig. 1a). The SEI impedance was fitted to a modified Randles circuit, where the semi-infinite Warburg impedance was replaced with a finite diffusion model featuring a reflective boundary (Fig. 1c, inset). This approach is equivalent to previous SEI fitting models[14,17,18], except that the finite diffusion transmission line used here better represents the multi-component nature of the SEI[19]. The $M_a$ element is used to approximate the transmission line element and calculate the ionic resistance of the SEI ($R_{SEI}$) using Equation S10 shown in Supplementary Note 3. The reflective boundary arises from the finite thickness of the SEI and the fact that the electrode remains above the Li plating potential. In total, 32 plating steps of 1.56 μA h cm$^{-2}$ each were performed. Figure 1d shows a progressive increase in resistance from finite diffusion ($R_{SEI}$) with increasing charge passed (see Supplementary Fig. S3 for the trend over time). However, this trend does not follow a linear relationship, as would be expected if the SEI grew uniformly in terms of composition and morphology. Instead, the resistance curve deviates to lower values. One possible explanation for this behaviour is the emergence of highly, yet not fully, lithiated Li$_x$P (where $x > 1.5$). This is reported to have a higher lithium diffusivity compared to Li$_3$P (Supplementary Fig. S4)[20]. If the initial 20 μA h cm$^{-2}$ of $R_{SEI}$ is fitted linearly through the origin-assuming negligible electronic conductivity and a fully dense SEI composed solely of

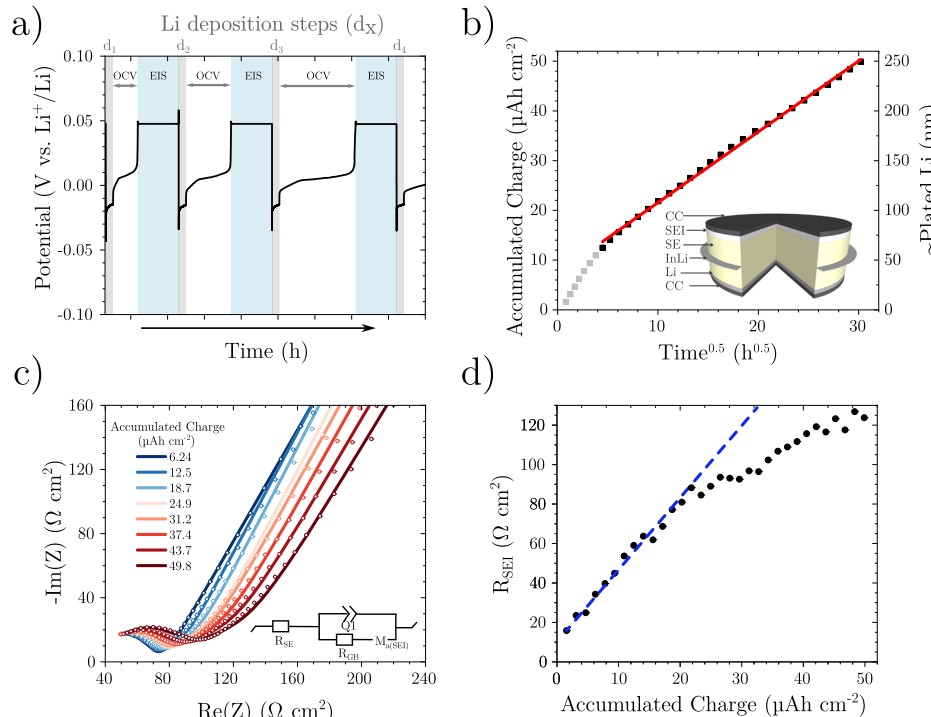

**Fig. 1 | Three-electrode CTTA. a** CTTA at 30 °C (15.6 μAcm$^{-2}$ for 6 min steps) with PEIS conducted once a 50 mV OCV was reached (see Supplementary Fig. S1 for real data), **b** relationship between accumulated charge and the square root of OCV time to consume Li metal (inset schematic of the three electrode setup), **c** PEIS of a growing SEI (circles data, lines fits) with the equivalent circuit (for details of the $M_a$ element see Supplementary Fig. S2), **d** calculated SEI resistance ($R_{SEI}$ from within $M_a$) before each titration step (with the straight dotted blue line indicating the nonlinearity of the results). Source data are provided as a Source Data file.

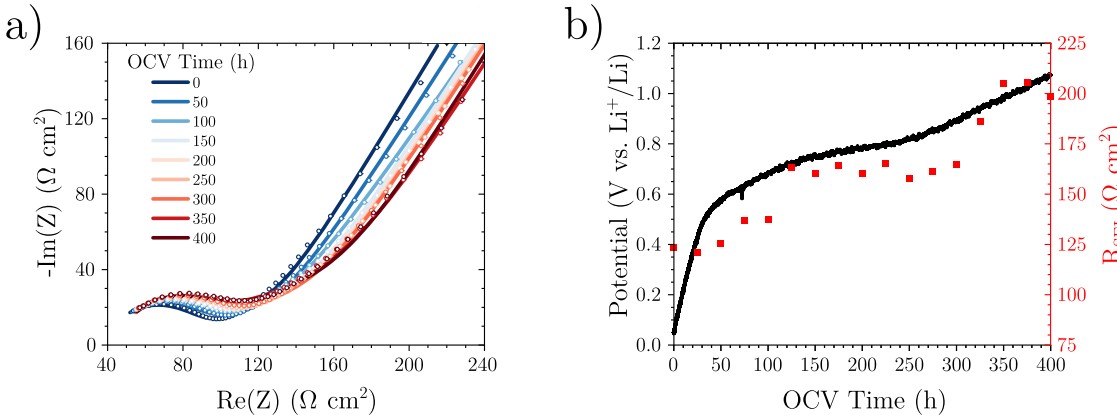

**Fig. 2 | Evolution of the SEI. a** Evolution of the SEI impedance in lithium-free conditions (circles data, lines fits) following on from CTTA from Fig. 2, **b** evolution of open-circuit voltage (OCV) and SEI resistance in lithium-free conditions. Source data are provided as a Source Data file.

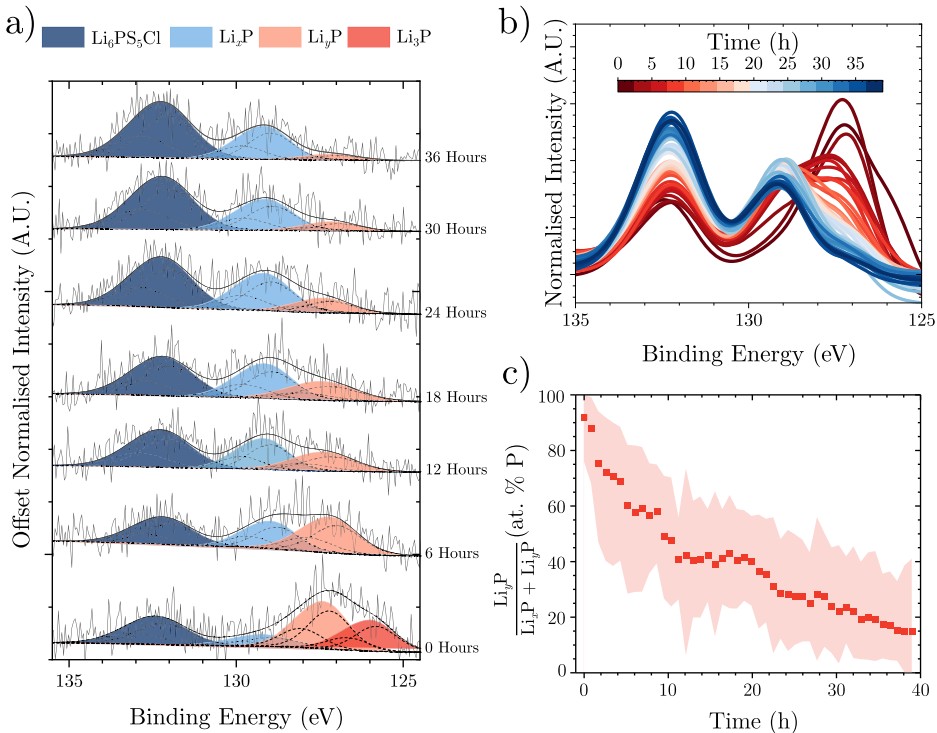

**Fig. 3 | Virtual-electrode XPS. a** XPS of phosphorus (maximum normalised) in the SEI after ~ 0.01 mAh cm$^{-2}$ of lithium was VEP through a Li$_6$PS$_5$Cl pellet over 1 h using an ebeam current of 2.5 μA (~ 0.01 mA cm$^{-2}$), **b** the mean normalised P *2p* fitted spectrum over time, **c** percentage of Li$_y$P in the P-containing SEI species (with the dots being the mean value and the shaded areas the uncertainty calculated from peak area standard deviations. 200 separate sets of random noise were added to the peak model envelope, and the peak model was refitted, enabling the standard deviation in peak area to be calculated). All P *2p* spectra are fitted as doublets (dotted lines), with shaded areas being summed for clarity and fitting details are provided in Supplementary Table 7. Source data are provided as a Source Data file.

Li$_3$P, Li$_2$S, and LiCl (where 1 μA h cm$^{-2}$ ~ 9 nm)[11],-an ionic conductivity of 204 (±8) nS cm$^{-1}$ is obtained. This value is directly comparable to the recently reported ionic conductivity of a synthetic SEI formed from Li$_6$PS$_5$Cl (134 nS cm$^{-1}$)[12].

After ~1000 h of CTTA and an accumulated charge of ~ 50 μA h cm$^{-2}$, the cell was left at open-circuit voltage (OCV), with PEIS measurements taken every hour for 400 h (Fig. 2a).

Over this period, the OCV was observed to increase, notably mirroring the delithiation profile of red phosphorus (Supplementary Fig. S5)[20]. Simultaneously, $R_{SEI}$ was seen to increase throughout the OCV period (Fig. 2b), indicating either SEI growth, a compositional change, or a combination of both.

To further investigate the cause of the change in OCV, VEP-XPS was performed[14]. Lithium (~ 0.01 mA h cm$^{-2}$) was plated through a 5 mm Li$_6$PS$_5$Cl pellet for 1 hour using an electron beam current of 2.5 μA (~ 0.01 mA cm$^2$), as previously reported[14].

XPS was then performed continuously, generating a new dataset every 53 min. Whilst Li$_2$S remains stable (Supplementary Fig. S6), the P *2p* spectra evolve over time (Fig. 3a–c). When freshly lithiated, the P *2p* spectra consist only of Li$_3$P, highly lithiated Li$_y$P and residual Li$_6$PS$_5$Cl. However, as the lithium metal is consumed, Li$_x$P emerges, accompanied by a loss of the Li$_3$P peak and a decrease in the Li$_y$P peak (0 < x < y < 3). These results indicate that the SEI composition evolves via the delithiation of phosphorus. In addition, an increase in

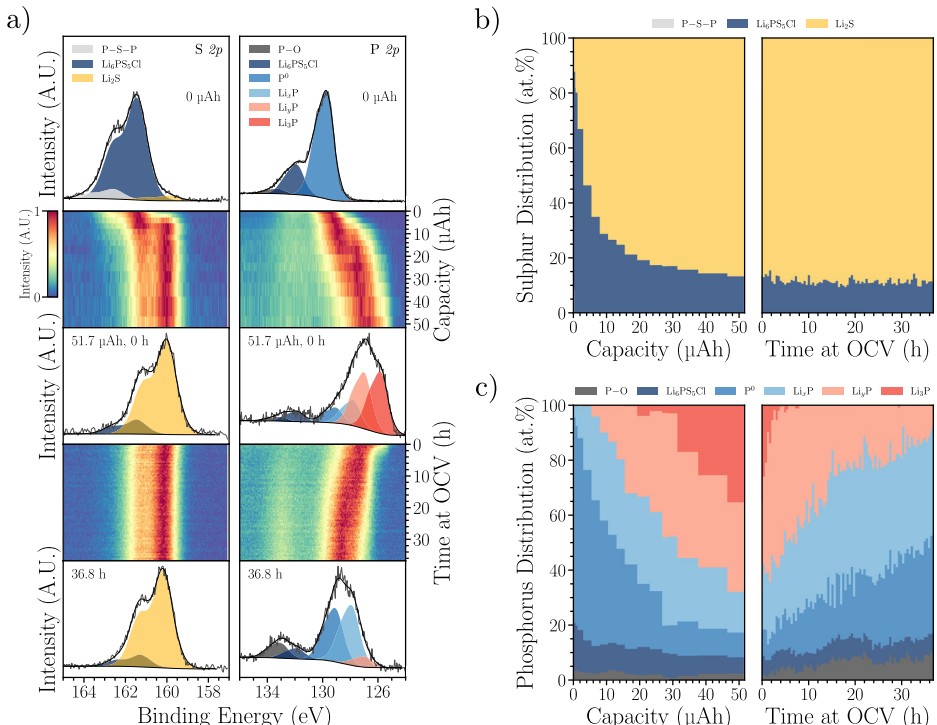

**Fig. 4 | Phosphorus lithiation. a** Evolution of the S *2p* and P *2p* spectra during VEP-XPS lithiation of a sputtered phosphorus layer on top of a $Li_6PS_5Cl$ pellet using an ebeam current of 30 μA to a capacity of 51.7 μAh (- 0.26 mAh cm$^{-2}$), followed by resting under OCV for 36.8 h. All spectra have been normalised such that intensities are scaled between 0 and 1 for clarity. **b** Breakdown of sulphur-containing species identified in (**a**). **c** Breakdown of phosphorus-containing species identified in (**a**). Fitting details are provided in Supplementary Table 8. Source data are provided as a Source Data file.

the $Li_6PS_5Cl$ peak is observed, likely due to the formation of a phosphate, which has been shown to be present in this binding energy region[12].

Due to the low signal-to-noise ratio of the phosphorus spectra, a second experiment was conducted. Here, phosphorus was directly sputtered onto the $Li_6PS_5Cl$ within the XPS analysis chamber, before being lithiated by VEP to a capacity of 51.7 μAh (-0.26 mAh cm$^{-2}$) and finally left at rest for 36.8 h (Fig. 4a). Prior to VEP ("0 μAh") the S *2p* spectrum is dominated by $Li_6PS_5Cl$ at 161.4 eV, with minor peaks attributable to P – S – P bonding and $Li_2S$, demonstrating that the sputtering process causes negligible electrolyte degradation. This is confirmed by the Cl *2p*, S *2p* and Li *1s* XPS spectra gathered during phosphorus sputtering in Supplementary Fig. 7, which show no shifts in binding energy. A corresponding $Li_6PS_5Cl$ peak is identified in the 0 μAh P *2p* spectrum at 131.8 eV, and the major peak at 129.7 eV is attributed to P$^0$. The minor peak at 133.2 eV is assigned to species containing P – O bonding.

During lithiation, the S *2p* $Li_2S$ peak grows in relative intensity while the $Li_6PS_5Cl$ peak reduces in intensity, indicative of electrolyte reduction. The proportion of $Li_2S$ increases rapidly from 4 at.% S prior to lithiation to 65 at.% S after 6.7 μAh, reaching 87 at.% S after 51.7 μAh (Fig. 4b). At the same time, the point of maximum intensity in the P *2p* region shift to lower binding energy and peak fitting after 51.7 μAh reveals three additional species at 127.7, 126.9 and 125.8 eV, attributed to $Li_xP$, $Li_yP$ and $Li_3P$, respectively, where $0 < x < y < 3$. The evolution of these phosphorus-containing species in Fig. 4c reveals that lithiation progresses through P$^0$ → $Li_xP$ → $Li_yP$ → $Li_3P$, and the proportions of the most lithiated phases increase with increasing capacity. Interestingly, significant electrolyte reduction was observed in the S *2p* spectra before $Li_yP$ and $Li_3P$ were detected, suggesting that even the least lithiated $Li_xP$ species can reduce $Li_6PS_5Cl$.

After lithiation, the sample was left to rest at OCV within the XPS analysis chamber. Figure 4a and b reveal negligible changes in the

sulphur-containing species during this period, suggesting $Li_2S$ is stable. On the other hand, during rest, the maximum intensity in the P *2p* spectra shifts back to higher binding energy, and the breakdown of phosphorus-containing species in Fig. 4c reveals that spontaneous delithiation occurs according to $Li_3P$ → $Li_yP$ → $Li_xP$ → P$^0$. The final oxidation of $Li_xP$ to P$^0$ must be accompanied by an additional reduction reaction, suggesting this delithiation process will lead to further electrolyte decomposition.

These observed trends in the P *2p* spectra are mirrored in the Li *1s* spectra in Supplementary Fig. 8, while no changes are observed in the Cl *2p* spectra. In the O *1s* spectra, it is evident that a $Li_2O$ peak at 528.4 eV emerges during lithiation and a peak at 531 eV grows during rest, indicative of the accumulation of surface oxygen species. While the P – O peak in Fig. 4a also grows in intensity during rest, it still remains only a minor phosphorus-containing species (Fig. 4c), so side reactions with gases inside the XPS chamber are not expected to have a significant impact on the observed phase evolution.

To delve deeper into the composition of the SEI, soft and hard X-ray photoelectron spectroscopy (SOXPES and HAXPES) measurements were performed. Photoelectrons emitted during SOXPES had a kinetic energy of 315 eV, and HAXPES was performed with incident photon energies of 2.2 keV and 6.6 keV, resulting in electron inelastic mean free paths (IMFPs) of ~1, ~5 and ~14 nm, respectively (Supplementary Fig. S9). SOXPES and HAXPES were first performed on a pristine $Li_6PS_5Cl$ pellet (Supplementary Fig. S10 and Supplementary Table 9), revealing only minor $Li_2S$, $Li_2SO_4$, $Li_2CO_3$ and LiOH impurities.

SOXPES and HAXPES were then performed following the in situ evaporation of approximately 20 nm of lithium metal onto the $Li_6PS_5Cl$ surface (denoted as 0 h) and then repeated after 6 and 12 h (Supplementary Fig. S11)[21]. Figure 5a shows the fitted S *2p* region where peaks attributed to both $Li_6PS_5Cl$ and $Li_2S$ can be observed and reliably separated. The S *2p* region can therefore provide information about

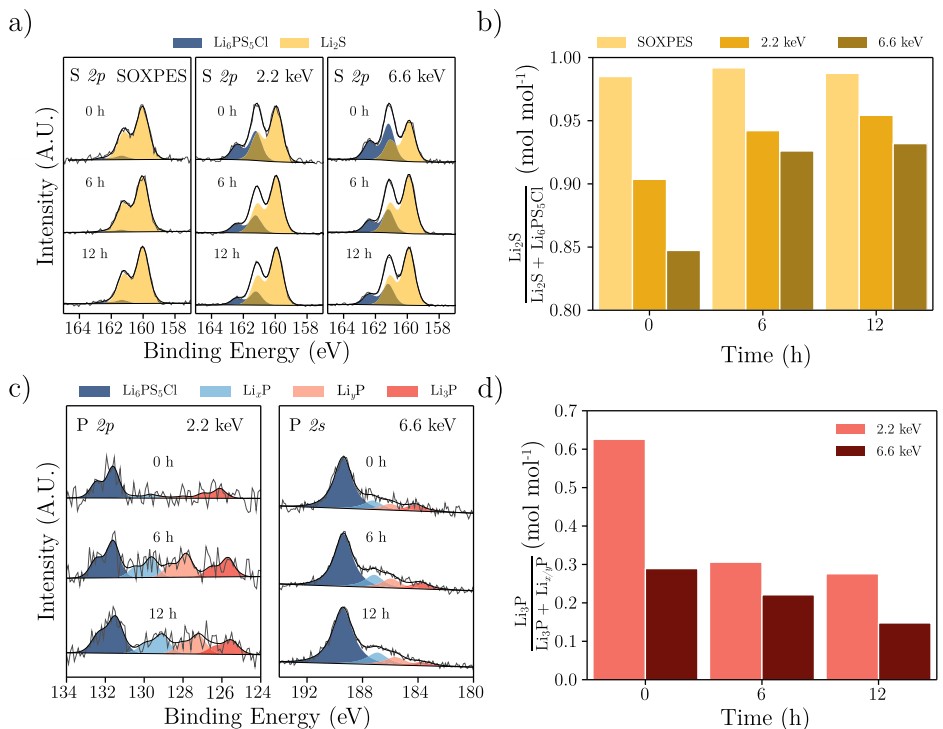

**Fig. 5 | SOXPES and HAXPES.** on an Li$_6$PS$_5$Cl pellet with ~20 nm of Li evaporated on top: **a** S $2p$ spectra gathered using SOXPES and HAXPES with time and incident X-ray energies marked on the plot. **b** Mole fraction of Li$_2$S in the S $2p$ region calculated from the spectra in (**a**), demonstrating an increase in the proportion of Li$_2$S with decreasing probing depth and with increasing time. **c** P spectra gathered using SOXPES and HAXPES with time and incident X-ray energies marked on the plot. **d** Mole fraction of Li$_3$P in the P-containing SEI species calculated from the spectra in (**c**), demonstrating a reduction in proportion of Li$_3$P with increasing probing depth and with increasing time. Fitting details are provided in Supplementary Table 10. All spectra are normalised such that intensities are scaled between 0 and 1 for clarity. Source data are provided as a Source Data file.

SEI formation and growth. At 0 h Li$_2$S is the dominant species in the SOXPES data, but its mole fraction decreases as the incident photon energy, and therefore probing depth, increases (Fig. 5b), enabling photoelectrons to be detected from the underlying Li$_6$PS$_5$Cl. Between 0 h and 6 h the SOXPES data remain almost constant, but the proportion of Li$_2$S evident at 2.2 and 6.6 keV increases, indicating growth of the SEI. The SEI then grows further between 6 and 12 h, although at a reduced rate. It is expected that the concentrations of LiCl and Li$_x$P will also increase in accordance with Li$_2$S, however, LiCl is indistinguishable from Li$_6$PS$_5$Cl, as there is no change in oxidation state. The different oxidation states of P are detectable, but the low concentration of P in Li$_6$PS$_5$Cl, combined with its low photoionisation cross section[22], unfortunately results in poorer signal-to-noise ratios for the corresponding P $2s$ and P $2p$ regions in Supplementary Fig. S11. Nevertheless, it is clear from the 2.2 keV P $2p$ region and particularly the 6.6 keV P $2s$ region in Fig. 5c that the SEI is composed of several P-containing species. Our binding energy assignments are consistent with previous reports[12], Li$_3$P at the lowest binding energy and Li$_x$P with decreasing lithium content as the binding energy increases. Figure 5d shows that the proportion of Li$_3$P in the SEI is highest at 0 h and small probing depth (2.2 keV), and this proportion reduces as both time and probing depth increase, allowing thicker SEIs to be studied. This suggests a layer of Li$_3$P forms first close to the lithium metal, and further SEI growth results in the formation of less lithiated Li$_x$P further away from the lithium negative electrode, resulting in a gradient of composition through the SEI. An additional experiment was performed to uncover how the SEI changes once the lithium metal is consumed. As evident in Supplementary Fig. S12, immediately after lithium evaporation, a gradient of lithiated P is again formed within the SEI, with less lithiated Li$_x$P observed further from the Li. However, after 22 h, once the lithium metal has fully reacted, the proportion of Li$_3$P decreases.

To gain further insights into the formation of Li$_x$P species in the SEI, cyclic voltammetry was performed with a planar stainless steel electrode setup (Fig. 6a). Stainless steel was chosen as it does not alloy with lithium and has negligible lithium diffusivity compared to other current collector materials[11].

During the first reductive sweep, two clear peaks can be observed (labelled $a$ and $b$ in Fig. 6a). Peak $a$ can be ascribed to the reduction of argyrodite. The onset potential of 1.13 V vs. Li$^+$/Li is below the computationally predicted value of 1.71 V vs. Li$^+$/Li[6], yet significantly higher than previously thought[23]. As reduction initially occurs above the theoretical full reduction potential of phosphorus to Li$_3$P (0.87 V vs. Li$^+$/Li), this peak can be assigned to the reduction of Li$_6$PS$_5$Cl to form Li$_2$S, LiCl and partially lithiated phosphorus species, which can be summarised as Li$_x$P, where $x$ < 3 (Equation (1)). Peak $b$ can be attributed to the lithiation of Li$_x$P to form the fully reduced Li$_3$P (Equation (2)). Indeed, fitting the areas of peaks $a$ and $b$ yields a capacity ratio of ~ 3:1 (Supplementary Fig. S14), which would be consistent with a conversion of LiP to Li$_3$P (Equations 1 and 2). The total capacity of the reductive sweep is 11.1 mC cm$^{-2}$, which would correspond to an SEI thickness of 28 nm (if fully dense and consisting of only Li$_2$S, LiCl and Li$_3$P).

$$\text{Li}_6\text{PS}_5\text{Cl} + (5+x)\text{Li}^+ + (5+x)\text{e}^- \rightarrow \text{Li}_x\text{P} + 5\text{Li}_2\text{S} + \text{LiCl} \qquad (1)$$

$$\text{Li}_x\text{P} + (3-x)\text{Li}^+ + (3-x)\text{e}^- \rightarrow \text{Li}_3\text{P} \qquad (2)$$

$$\text{Li}_3\text{P} \rightarrow \text{Li}_x\text{P} + (3-x)\text{Li}^+ + (3-x)\text{e}^- \qquad (3)$$

During the oxidative sweep, a peak can be seen (labelled $c$) at 0.87 V vs. Li$^+$/Li, in line with the reoxidation of Li$_3$P to Li$_x$P

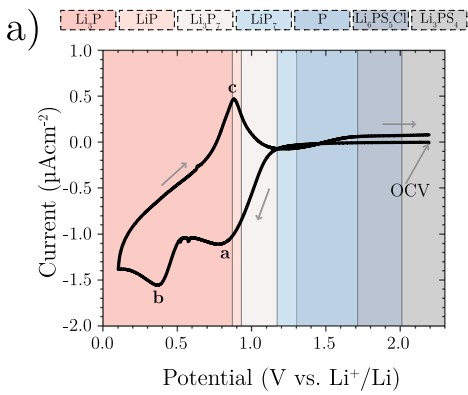

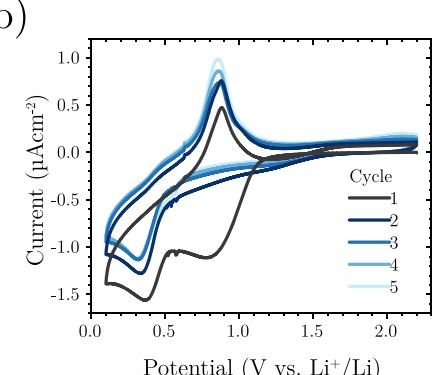

**Fig. 6 | Cyclic voltammetry of Li₆PS₅Cl to determine the reduction potential.** A Li₆PS₅Cl pellet was placed between a stainless steel working electrode and a Li counter. The scan rate was 100 μV s⁻¹ and the starting potential 2.2 V vs. Li⁺/Li, whilst a uniaxial applied pressure of 13 MPa was used. **a** First cycle voltammogram.

The background colouring represents the theoretical stability windows of phosphorus species according to Zhu et al. (for all species see Supplementary Fig. S13)[6]. **b** Additional cycles. Source data are provided as a Source Data file.

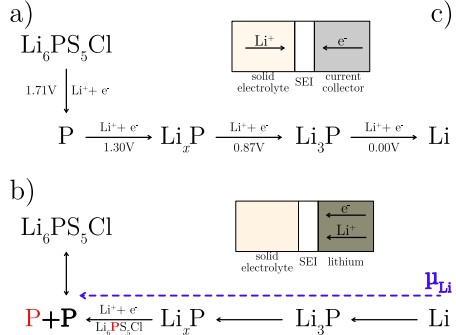

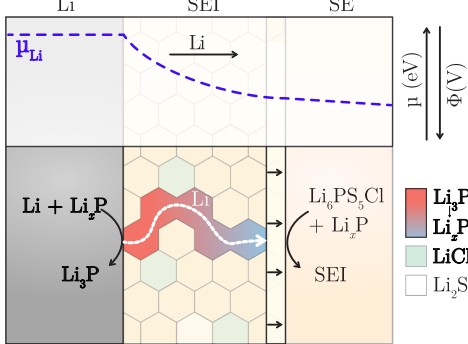

**Fig. 7 | Mechanism of SEI growth. a** Decomposition pathway of Li₆PS₅Cl, highlighting the importance of partially lithiated Li$_x$P species in the growth of the SEI. **b** The lithium pathway through phosphorus, enabling a growing SEI at OCV

conditions. **c** The chemical potential gradient over the SEI resulting in a gradient of phosphorus lithiation and the resulting SEI growth reactions.

(Equation (3))[6,20,24]. Interestingly, this occurs at a potential that is below the observed initial Li₆PS₅Cl reduction potential. On subsequent cycles (Fig. 6b) peak *a* is no longer clearly visible suggesting that most of the SEI growth occurs on the first reductive sweep. Nonetheless, an increase in the peak current of peak *c* with cycling is indicative of a continuous growth of the SEI.

## Discussion

The experimental evidence suggests the presence of a gradient of lithiated phosphorus in the SEI. This is in agreement with a recent study by Ren et al. which computationally predicted that the SEI of Li₃PS₄ contains at least two phosphorus regions[25]. One region with a Li to P coordination number of 11 near the Li electrode, corresponding to Li₃P, and another region with a Li to P coordination number of 6 near the solid electrolyte, corresponding to LiP. However, previous HAXPES research has failed to see this gradient due to an inability to observe phosphorus in the SEI due to a Ni current collector and lithium oxide attenuating the beam[26], while previous VEP-XPS measurements have failed to observe the instability of lithiated phosphorus with the solid electrolyte due to either a limited observation time or a different SE being studied[14,27].

The gradient of lithiated phosphorus in the SEI can be explained by considering the formation of the SEI in a lithium-free configuration (as illustrated in Fig. 7a). During the first charge, the potential of the stainless steel current collector decreases. Once it falls below the theoretical electrochemical stability threshold of Li₆PS₅Cl (1.71 V vs. Li⁺/Li), Li₆PS₅Cl undergoes reduction, forming Li₂S, LiCl, and P, all of

which remain in equilibrium with Li₆PS₅Cl. As the electrode potential decreases further, falling below 1.30 V vs. Li⁺/Li, P begins to lithiate, initially forming partially lithiated Li$_x$P species.

These reactions require electrons from the current collector and ions from the solid electrolyte side to proceed. As both are readily available, the initial growth of the SEI is likely kinetically limited. However, as the SEI thickness increases, mass transport becomes the limiting factor. This is consistent with the CV data in Fig. 6a, where peak *a* corresponds to the kinetically driven decomposition of Li₆PS₅Cl to form Li$_x$P species, followed by a mass transport-limited regime as the SEI thickens. Determining whether electron or ion transport is the primary limiting factor would require further investigation.

At 0.87 V vs. Li⁺/Li, full lithiation occurs, yielding Li₃P. This is again in agreement with what is observed in Fig. 6a, where peak *b* corresponds to the lithiation of Li$_x$P in the SEI to form Li₃P. Li₃P is a mixed electron-ion conductor, and it is therefore likely that at this potential, the SEI is mostly composed of fully reduced Li₃P. This process is reversible, as shown by peak *c* in Fig. 6a corresponding to the delithiation of Li₃P. At this stage, all SEI components-Li₃P, LiCl, and Li₂S-are thermodynamically stable down to the lithium plating potential. Consequently, at 0 V vs. Li⁺/Li, metallic lithium begins to plate onto the current collector.

After charging and while at open-circuit potential, chemically driven decomposition occurs (Fig. 7b). On the solid electrolyte front, Li₃P reacts with fresh argyrodite, forming new SEI while undergoing delithiation. This process establishes a lithium chemical potential gradient within the SEI, driving coupled Li₊/e⁻ diffusion from the

lithium front through the SEI to the solid electrolyte front. This would explain the experimentally observed lithium gradient (Li $\rightarrow$ $Li_3P$ $\rightarrow$ $Li_xP$ $\rightarrow$ P) and the diffusion-limited growth of the SEI (see Equations S1-6 for stoichiometric reactions).

Whilst the Janek group has interpreted the linear dependence of SEI growth as a function of $t^{0.5}$ they observed using a Wagner model for diffusion-controlled solid-state reactions[9,11], our results clearly show a non-linear dependence on $t^{0.5}$ during the first 20 h (Fig. 1b and Supplementary Fig. S15). We have therefore explored the Deal-Grove model, originally developed to describe the thermal oxidation of silicon, and previously applied to the chemical growth of the SEI in the liquid state, to interpret our data[28,29]. This model includes an initial surface reaction-controlled growth, which follows a linear dependence on time, and a diffusion-limited parabolic growth regime, which dominates at longer times. However, this model does not fully capture the growth mechanism either, as it not only provides a poor fit to the experimental data but also fails to describe the initial SEI growth, which is not linear with time. Nonetheless, fitting the SEI growth at long times predicts that the kinetically limited stage of SEI growth is restricted to < 3 µA cm$^{-2}$ (see Supplementary Note 2), which is in good agreement with the charge under peak $a$ in Fig. 6a. We therefore propose that SEI growth is indeed diffusion-limited, but that the diffusivity is not constant. Instead, it is a function of SEI composition, which-as described previously-varies over both time and thickness. This variation in diffusivity could be due to the variation of diffusivity of Li in P with lithiation state[20], the change of the phosphorus volume fraction in the SEI through lithiation states (Supplementary Table S1, 2), or an evolving porosity. Further investigations are required to validate this mechanism.

Two important points should be noted. First, this process relies on the presence of a phosphorus percolation pathway throughout the SEI, highlighting the critical role of SEI nanostructure. Indeed, changing the chemical composition of argyrodite without significantly changing the volume fraction of phosphorus in the SEI (Supplementary Table S1–4) showed minimal changes to the kinetics of SEI growth[30]. While the ability to lithiate phosphorus in the SEI was found to be key to the difference in growth kinetics between $Li_{10}GeP_2S_{12}$ (LGPS) and $Li_{1.5}Al_{0.5}Ge_{1.5}(PO_4)_3$ (LAGP)[31]. Meanwhile, the graded structure of the SEI, which ensures that the most reduced phosphorus species are not in physical contact with the solid electrolyte, has been linked to the superior stability of the LiPON SEI[32]. Second, if a continuous percolation pathway is present, SEI growth will persist as long as metallic lithium is available. Once all the plated lithium is consumed, the remaining $Li_3P$ will begin to delithiate, ultimately leaving an SEI composed solely of LiCl, $Li_2S$, and P.

Although this work has only looked at the solid electrolyte $Li_6PS_5Cl$, we believe that this continuous mechanism of SEI formation would be present in any solid electrolyte that can be reduced to contain partially lithiated phosphorus, or potentially any element that can alloy with lithium. Indeed, a loss of $Li_3P$ in SEI at rest has been observed over time with $Li_7P_3S_{11}$[9]. While this observation was speculated to be due to reactions with trace amounts of oxygen or water in the UHV XPS chamber, our results indicate that it may be exhibiting SEI evolution similar to that of the SEI of $Li_6PS_5Cl$. More recently, El Kazzi's group has shown multiple degrees of lithiation of phosphorus in the SEI of $Li_3PS_4$[33], indicating that phosphorus may play the same role in its SEI.

The practical implications of this are quite impactful to the implementation of solid-state batteries, as the SEI growth leads to impedance growth as well as lithium and solid electrolyte consumption. Work by the Janek group has predicted the need to keep SEI resistance below 10 $\Omega$ cm$^2$[12,34], highlighting the importance of finding a mechanism to halt diffusion-controlled SEI growth. By understanding the diffusion mechanism behind this growth, potential routes to stop it can now be explored. We do not expect metallic interlayers to help, but

electronically insulating interlayers could, by introducing a steep decrease in the electrochemical potential of the electrons from the current collector interface to the SE interface, so at the SE interface, the electrons are below the electrolyte decomposition potential. Supplementary Fig. S16 reveals that a $Li_2O$ interphase slows the rate of SEI growth by over a factor of 5, compared to a traditional reduced $Li_6PS_5Cl$ interphase. It shouldn't therefore come as a surprise that the impedance using lithium metal foil negative electrodes seems to grow less rapidly than in the Li-less configuration[17], as lithium metal foils have a native passivation layer. Li-less approaches might be more challenging to implement.

In conclusion, SOXPES and HAXPES studies revealed a phosphorus lithiation gradient in the SEI, and VEP-XPS confirmed that $Li_3P$ in the SEI is not stable and will undergo spontaneous delithiation to $Li_xP$, whilst voltammetry revealed that $Li_6PS_5Cl$ can be reduced above the oxidation potential of $Li_3P$. These results show that without a phosphorus-free passivating interlayer between the Li negative electrode and argyrodite, the SEI will continually grow due to the reaction between lithiated phosphorus and $Li_6PS_5Cl$ and subsequent diffusion of lithium through the SEI. The growing SEI simultaneously reduces the coulombic efficiency and energy density of solid-state cells, whilst increasing the overall cell impedance. Assuming a fully dense SEI, the conductivity is seen to grow non-linearly, initially at a rate of ~204 nS cm$^{-1}$ before increasing.

## Methods

### Electrochemistry

Electrochemistry experiments were performed at a pressure of 13 MPa in an argon chamber at an environmental temperature of 30 °C using a VMP3 BioLogic potentiostat. All two-electrode experiments used 100 mg of $Li_6PS_5Cl$ (Ampcera 10 µm, synthesised from > 99.9% precursor materials) pressed at 370 MPa for 300 s using stainless steel plungers in a Macor cylinder (internal diameter = 10 mm), to yield a pellet of ~700 µm in thickness. Lithium (99.9% Alfa Aesar) counter electrodes were prepared by brushing and calendering lithium to 200 µm, cutting out a 10 mm disc and pressing onto the bottom side of a $Li_6PS_5Cl$ pellet at 80 MPa for 30 s. Stainless steel working electrodes were prepared by simply placing a stainless steel plunger directly onto the top side of the $Li_6PS_5Cl$ pellet inside the Macor cylinder. For three-electrode measurements, a PEEK cylinder was used in place of the Macor cylinder, which was cut in half perpendicular to the internal 10 mm hole and held together with 3 screws. Between the two pieces of PEEK, an InLi-In alloy ring (internal diameter 10 mm) was placed, then aligned with a stainless steel plunger. A fourth screw was used to make electrical contact to the InLi-In ring. The cut of the PEEK body was positioned so that the InLi-In alloy ring was positioned in the middle of a 150 mg $Li_6PS_5Cl$ pellet. All other cell assembly steps were identical to two electrode cell assemblies. All cells were left in an environmental chamber for a minimum of 10 h prior to experiments commencing, to allow for temperature stabilisation.

All voltammetry experiments were conducted at a scan rate of 100 µV s$^{-1}$ and a starting potential of 2.2 V vs. Li$^+$/Li. Cyclic voltammetry used a first vertex potential of 0.1 V vs. Li$^+$/Li and a second vertex potential of 2.2 V vs. Li$^+$/Li. CTTA applied a current of − 12.25 µA for 6 min during the plating steps and a 50 mV vs. Li$^+$/Li cut-off potential. Once 50 mV vs. Li$^+$/Li was reached, potentiostatic electrochemical impedance spectroscopy (PEIS) was conducted, before the next plating step. PEIS was conducted between frequencies of 400 kHz to 10 Hz with 10 points per decade, a logarithmic spacing and two measurements taken per frequency. A sinus amplitude of 10 mV was used, and a 0.1 s wait period was used before each frequency and drift correction was used. Impedance spectra were fitted using the built-in Z fit function inside EC-lab. A Randles circuit was used, with the Warburg element replaced by an open-circuited uniform distributed resistor constant phase element (URQ) transmission line (Supplementary

Fig. S2). All fitting data not shown in the main text can be found in Supplementary Table S5 and S6 along with Supplementary Fig. S17.

An InLi-In alloy was used as a reference in 3-electrode cells, due to creep issues when Li rings were tested. The alloy was prepared as follows. Indium (Alfa Aesar 99.99%, lump) and lithium (99.9% Alfa Aesar) were weighed (MTI PCB-200) to make a 75:25 atomic ratio. These were put into a lined (molybdenum foil, Goodfellow) stainless steel crucible. The mixture was heated to 700 °C in a box furnace (MTI KSL-1200X-J-UL) within an argon glovebox. Once molten, the crucible was removed from the furnace, stirred with a spatula and returned to the furnace for 2 h, then the crucible was removed, and the alloy melt poured onto the stainless-steel glovebox base such that it rapidly cooled, leading to a fine two-phase microstructure.[35] Physically, the cast alloy looked like indium and was soft. Characterisation of the cast two-phase alloy is provided in the supplementary information of Aspinall et al.[16]. The electrochemical potential of this cast alloy was found to be stable within a fraction of a millivolt at 0.622 V vs Li$^+$/Li.

The reference ring (Fig. 2b insert) minimises impedance artifacts as the SE separator, working electrode and counter electrode all have the same diameter and are aligned to each other in the PEEK housing, whilst the hole in the ring matches the SE separator diameter[36]. The ring design allowed the reference to be directly next to the field lines, yet not impede the flow of Li ions as previous three electrode solid state setups have done[37–40].

### X-Ray Photoemission Spectroscopy

For XPS, a 5 mm diameter argyrodite pellet (30 mg) was placed on top of a 4.76 mm diameter Li foil, which was then placed on top of a copper foil. This stack was pressed at ~50 MPa to ensure good contact. The sample stack was grounded to the XPS sample stage using carbon tape. The sample stage was then transferred from the Ar glovebox to the XPS system (PHI VersaProbe III) using a vacuum transfer vessel (MOD 07-111K, ULVAC-PHI, Inc.) to avoid air contamination. The X-ray source was Al Kα (hν = 1486.8 eV) and the vacuum level maintained below 10$^{-6}$ Pa throughout this experiment. After collecting the XPS data on the pristine surface, the pellet was exposed to a ~5 mm electron beam for 1 h from an electron neutraliser within the XPS system, driving lithium-ion migration through the pellet. The current of the applied electron beam was set as 2.5 μA, corresponding to a current density of 12.74 μA cm$^{-2}$. After the electron beam exposure, XPS measurements were performed continuously. Each XPS measurement was conducted as follows: a survey scan (pass energy of 224 eV), then high-resolution scans (pass energy of 55 eV) of P 2p, Li 1s, S 2p, O 1s, Cl 2p, and C 1s. Each full sequence of measurements took ~53 min to complete.

For the phosphorus lithiation experiment, the argyrodite-Li stack was prepared as above, and a phosphorus chunk (Sigma-Aldrich, 99.99% trace metals basis) with approximate dimensions of 5 mm × 6 mm was affixed to the target holder of the XPS stage. Phosphorus was sputtered onto the solid electrolyte surface inside the XPS analysis chamber by bombarding the phosphorus target with an Ar$^+$ gun with an accelerating voltage of 4 kV for a total of 2 h (Supplementary Fig. 7). The sample surface was then alternately exposed to a 30 μA electron beam and analysed until peaks consistent with fully-reduced Li$_3$P and Li$^0$ were detected, corresponding to a capacity of 51.7 μAh. VEP steps were then halted, and further XPS measurements were performed continuously while the sample was allowed to rest at open-circuit. Each measurement series took ~27 min.

Soft and hard X-ray photoelectron spectroscopy (SOXPES and HAXPES) was performed at the I09 beamline at the Diamond Light Source (Didcot, UK). SOXPES was performed with variable incident X-ray energies, providing emitted photoelectrons with a kinetic energy of 315 eV, while HAXPES was performed with incident X-ray energies of 2.2 and 6.6 keV. This results in electron inelastic-mean-free-paths (IMFPs) of approximately 1, 5 and 14 nm for the SOXPES, 2.2 keV and

6.6 keV photons, respectively (Supplementary Fig. S9). IMFPs were estimated using the TPP-2M formula for both Li$^0$ and Li$_6$PS$_5$Cl, giving both upper and lower bounds, respectively. To minimise beam damage, the undulator was detuned to reduce the incident photon intensity by factors of approximately 10 and 100 for HAXPES and SOXPES, respectively. This resulted in a beam size of 300 μm × 300 μm. The angle between the incident beam and the sample surface was 15°, giving an analysis area of 300 μm × 1200 μm. Photoelectrons were detected using a concentric hemispherical analyser (VG Scienta EW4000, ± 28° lens acceptance angle) operated with pass energies of 200 and 100 eV for HAXPES and SOXPES, respectively. Li$_6$PS$_5$Cl pellets were prepared as above and introduced into the end station using an inert transfer vessel. A 20 nm thick film of lithium metal was thermally evaporated onto the Li$_6$PS$_5$Cl in situ using a custom-built ultrahigh vacuum (UHV) chamber attached to the beamline end station[21], and all samples were stored under UHV throughout the duration of the experiment.

All collected data was processed using CasaXPS software[41], and calibrated to the Cl 2p$_{3/2}$ peak at 198.5 eV. The standard deviation in the individual P 2p peak areas in Fig. 3c was estimated using CasaXPS and used to calculate the uncertainties presented in Fig. 3e. The mole fraction of Li$_2$S in the S 2p region, $x_{Li_2S}$, in Fig. 5b was calculated from the areas of the Li$_2$S peak, $A_{Li_2S}$, and the Li$_6$PS$_5$Cl peak, $A_{Li_6PS_5Cl}$, according to Eq. (4):

$$x_{Li_2S} = \frac{A_{Li_2S}}{A_{Li_2S} + \frac{1}{5}A_{Li_6PS_5Cl}} \tag{4}$$

## Data availability
All data generated in this study have been deposited in the Zenodo database under a Creative Commons Attribution 4.0 International License (https://doi.org/10.5281/zenodo.15849598). Source data are provided with this paper in the Source Data file. Source data are provided in this paper.

## Code availability
All data created during this research are openly available from the data archive at https://doi.org/10.5281/zenodo.15849598.

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

## Acknowledgements

This work was supported by the Faraday Institution SOLBAT and Characterisation projects (grant numbers FIRG056, FIRG020 and FIRG087) and Henry Royce Institute (through UK Engineering and Physical Science Research Council grant EP/R010145/1). M.P. is grateful for the support of Nissan Motor Co., Ltd., Japan. B.J is grateful for the support of the Clarendon Fund Scholarships. R.S.W. acknowledges a UKRI Future Leaders Fellowship (MR/V024558/1) and funding from the European Research Council (ERC) under the European Union's Horizon 2020 research and innovation programme (EXISTAR, grant agreement No. 950598). We acknowledge Diamond Light Source for synchrotron beamtime on the I09 beamline (SI 25807) and thank the beamline staff for their support. Tien-lin Lee is also acknowledged for helping develop the synchrotron Li deposition setup and helping conduct some SOXPES/HAXPES experiments.

## Author contributions

M.B. and M.P. conceived the idea. M.B. performed all the experiments, with the exception of XPS, which was performed by Y.L., whilst J.A. made the InLi-In alloy. SOXPES and HAXPES measurements were conceived and by J.S.G. and R.S.W. and performed by J.E.N.S., J.S.G. and R.S.W., Z.L. conducted the Li2O interphase experiment. Y.L. and B.J. fitted the XPS spectra, and B.J. analysed the SOXPES and HAXPES data with input from J.S.G. and R.S.W. M.B. wrote the manuscript with input from all authors. M.P. supervised the design of the project and provided frequent input on the interpretation of all results.

## Competing interests

The authors declare no competing interests.
