## [Transparent Peer Review file · Nature Communications]

The Role of Phosphorus in the Solid Electrolyte Interphase of Argyrodite Solid Electrolytes

Corresponding Author: Professor Mauro Pasta

Version 0:

Reviewer comments:

Reviewer #1

(Remarks to the Author)

In this study, the authors investigate the composition, thickness, and stability of the solid electrolyte interphase (SEI) layer formed when plating metallic lithium on an argyrodite Li₆PS₅Cl (LPSCI) solid electrolyte pellet. They employ a combination of electrochemical techniques, including coulometric titration time analysis (CTTA) and potentiostatic electrochemical impedance spectroscopy (PEIS), along with surface-sensitive X-ray photoelectron spectroscopy (XPS) to analyze a 20 nm thick lithium layer over time. The findings reveal that the SEI consists of Li₂S, LiCl, and Li_xP, with a gradient of Li_xP present. The authors suggest that highly reduced species, such as Li₃P, form near the metallic lithium interface, while delithiated Li_xP species develop in contact with LPSCI. A continuous growth of the SEI is attributed to the reaction between LPSCI and Li₃P to form Li_xP, which persists as long as phosphorus percolation pathways remain in the SEI. The study emphasizes the importance of accurately identifying SEI species and understanding their formation and stability. However, some of the following aspects require further clarification.

(1) Introduction: Please reformulate that lithium non-uniformity in liquid-based batteries can be circumvented by the application of solid electrolytes due to their mechanical strength. This formulation appears misleading to me, as it leaves out that this is only possible with homogeneous stack pressure application. Without it, solid-state batteries also suffer inhomogeneous Li-plating.

(2) Introduction: Is it really just the “low” ionic conductivity of oxide solid electrolytes, or also their bad interface contact that plays a major role in the challenges related to their commercialization (which relates back to their mechanical properties)? Also this formulation appears misleading to me.

(3) In Figure 1c, where do you fit your R_{SEI}? It is not clear to me from which resistance in your equivalent circuit you derive the R_{SEI} values, that you plot in Figure 1d.

(4) How do the authors translate their observations of continuous SEI growth to the scenario of “zero-lithium-excess” solid-state batteries? In a real setup of “zero-lithium-excess”, we would not have a continuous reaction front between metallic lithium and LPSCI. On the contrary, as we would form species of less reduced phosphorus (P, Li_xP) first between LPSCI and the current collector (see CV in Figure 5a), and after that plate lithium. Would that impact the stability of the SEI and limit its continuous evolution?

(5) The low signal-to-noise ratio on the P2p spectra in Figure 3a and 4c makes it difficult to conclude on the different Li_xP species and their accurate binding energy positions. Did you calculate the error that P2p induces in your quantitative analysis in Figure 4d?

(6) Please list your XPS fitting parameters for S2p, P2p and P2s in the supporting information (full width at half maximum and binding energy positions of the fitted compounds).

(7) How do you explain your Li1s spectra after 22h in Figure S11a? The component that you previously assigned to Li₂O is shifted and split in two components now – could that indicate other species forming after such a long time instead of Li being only consumed by SEI growth?

(8) Your assignment of the peaks in the CV of Figure 5a is not clear to me. If I understood correctly, you assign peak “a” to the formation of Li_xP, and peak “b” to their lithiation until Li₃P. I think peak “a” should be the formation of Li₂S and partially P, and peak “b” should be Li_xP to Li₃P formation, which is reversed on peak “c”. As there is no further peak upon oxidation, and in subsequent reduction peak “a” has disappeared, it would make sense to correlate peak “a” to the formation of Li₂S, which is not reversed at such low potentials.

(9) What about the reversibility of such SEI layer? From CV, I can tell that the phosphorous species form reversibly. Did the authors study with CTGA, EIS and XPS (or other techniques) also the reversibility of such SEI growth or could elaborate on

the implications on the SEI composition and stability upon subsequent battery cycles?

(10) Minor side note on typos: Please always leave the reference [x] either before or after your punctuation, it appears to be mixed sometimes. Figure S5, please indicate which is your data and what is the reference in the labels for clarity.

Reviewer #2

(Remarks to the Author)

This study aims to understand the role of phosphorus in the diffusion-controlled growth of the interphase using electrochemical and X-ray photoelectron spectroscopy techniques. While this topic is of significant interest in sulfide-based all-solid-state batteries, neither the findings nor the employed techniques appear to be novel. The methodology closely resembles that of a study reported by the Janek group (Nat Commun 14, 6946 (2023)), and the findings have already been presented by the same research group in a previous work (Nat Commun 13, 7237 (2022)). Additionally, similar conclusions were drawn in ACS Energy Lett. 2024, 9, 7, 3492–3500, where hard X-ray photoelectron spectroscopy (HAXPES) was used. Furthermore, although not specifically focused on Li₆PS₅Cl, the evolution of the solid electrolyte interphase and its chemical composition has been investigated using operando X-ray photoelectron spectroscopy in Nat Commun 9, 2490 (2018). Given these considerations, I doubt whether this study meets the standards expected for publication in Nature Communications.

Reviewer #3

(Remarks to the Author)

In this manuscript, the authors aim to elucidate the growth mechanism of the solid electrolyte interphase (SEI) formed at the interface between the sulfide-based solid electrolyte Li₆PS₅Cl and lithium metal. To achieve this, they employed a combination of techniques including electrochemical measurements, X-ray photoelectron spectroscopy (XPS), and both hard and soft X-ray photoelectron spectroscopy (HAXPES and SOXPES). Particular focus was placed on the presence of lithium phosphide species (Li₃P) within the SEI and their potential redox behavior during SEI formation.

The experimental approach is systematic and the methodologies used are appropriate. However, the findings largely overlap with previously reported results and do not offer substantial new insights. More critically, the photoelectron spectroscopy analysis—which constitutes the central component of this work—contains fundamental errors in both its interpretation and application of the technique. These issues require careful reevaluation from the level of basic principles.

For example, the core-level P 2p spectra were fitted using a single peak, despite the well-established fact that 2p orbitals must be analyzed as a spin-orbit doublet. There are no exceptions to this requirement.

In light of these issues, I regret to conclude that this manuscript is not suitable for publication in its current form.

Version 1:

Reviewer comments:

Reviewer #1

(Remarks to the Author)

First, I wish to thank the authors for addressing my comments and questions accordingly and in great detail. They have enhanced the clarity and significance of the manuscript, especially with additional experiments on the lithiated phosphorus evolution using VEP-XPS.

I have three comments concerning the author's changes and answers to my previous review:

- (1) For clarity, please bring component P-S-P in Figure 4a to the front, or the color-code is not clear anymore.
- (2) Please stay consistent with the labelling for the lithiated phosphorus species. In Figure 4 they are split in Li₃P, Li₂P and Li₃P, but in Figure 5 they are summarized as Li₃P and Li₃P, though the fitting indicates three species.
- (3) Thank you for the clarification on Figure 6. Indeed, it is confusing to read the manuscript description and to look at different species highlighted in Figure 6, which come from theoretical predictions. If the authors agree, I would suggest unifying the species assignment displayed in Figure 6 with the description provided below the graph.

Reviewer #2

(Remarks to the Author)

The authors have made a commendable effort to address previously reported studies, including those, I highlighted as similar, and have provided convincing arguments to underscore the novelty of their work. However, there remain several points that require further clarification and deeper investigation to strengthen the overall impact and scientific rigor of the study:

1. The current study focuses on an argyrodite-type sulfide solid electrolyte. Would the same interphase growth mechanism and SEI evolution behavior be expected for other classes of sulfide-based electrolytes? In particular, I am curious about how the role of phosphorus in the SEI might vary depending on the structural nature of the anionic units, such as ortho-thiophosphates (PS₄³⁻), pyro-thiophosphates (P₂S₇⁴⁻), hypo-thiophosphates (P₂S₆⁴⁻), and meta-thiophosphates (P₂S₆²⁻). While I understand the scope of this work is centered on a specific electrolyte type, a brief discussion of how the findings might generalize to other chemistries would enhance the broader relevance of the study.
2. The use of distribution of relaxation times (DRT) analysis would significantly enhance the interpretation of the EIS Nyquist

plots. I strongly encourage the authors to perform DRT analysis and relate the resulting features to the evolution of the SEI. This would provide deeper insight into the individual EIS plots contributing to the interfacial processes.

3. The manuscript suggests that lithium diffusion through a percolated LixP phase is the rate-limiting step. However, this assumption relies on an unverified comparison of the diffusivities of LixP and Li3P. Could the observed non-linear growth behavior instead (or additionally) stem from evolving SEI porosity, phase morphology, or changes in percolation connectivity? A more quantitative or at least qualitative discussion on this aspect would strengthen the proposed mechanism.

4. The reversible delithiation observed via VEP-XPS and voltammetry is a central part of the proposed instability mechanism. However, its implications for long-term cycling behavior are not fully addressed. How reversible is this delithiation beyond the immediate OCV rest period? Does this process result in partial dissolution or dynamic reformation of the SEI over extended cycling? Clarifying this would be important for understanding the stability and durability of the interphase.

5. All experiments in the manuscript appear to have been conducted under a stack pressure of 13 MPa. Have the authors considered conducting similar tests under lower or variable pressures to assess the sensitivity of SEI evolution to mechanical loading? Such data would greatly enhance the relevance of this work for practical battery integration, where high stack pressures may not always be feasible.

Reviewer #3

(Remarks to the Author)

This manuscript is worthy of publication once the remaining revisions requested by the editorial office have been addressed.

Version 2:

Reviewer comments:

Reviewer #2

(Remarks to the Author)

The authors have addressed the concerns. It is now suitable for publication.

Response to the reviewers comments

First, we would like to thank all the reviewers for taking the time to review our manuscript.

Reviewers 2 and 3 raise concerns regarding the novelty of our work, and we would like to address this point upfront.

While the phenomenon of diffusion-limited SEI growth has indeed been observed in prior work, the underlying mechanism responsible for this behaviour had not, to our knowledge, been identified. In our study, we provide direct operando evidence using both EIS and in-situ XPS, two techniques we consider uniquely suited for this purpose, for the existence of a phosphorus gradient within the SEI. Based on these observations, we propose a mechanism in which phosphorus plays a critical role in the growth dynamics of the SEI.

We believe that identifying and substantiating this mechanism is a significant advance, as it not only clarifies a previously unexplained phenomenon but also points toward potential strategies for mitigating it. We respectfully submit that this mechanistic insight constitutes a significant and novel contribution that extends beyond previously reported observations. Reviewer 1 aptly captures the essence of our paper in stating *“The study emphasizes the importance of accurately identifying SEI species and understanding their formation and stability.”*

Reviewer #1

In this study, the authors investigate the composition, thickness, and stability of the solid electrolyte interphase (SEI) layer formed when plating metallic lithium on an argyrodite Li₆PS₅Cl (LPSCl) solid electrolyte pellet. They employ a combination of electrochemical techniques, including coulometric titration time analysis (CTTA) and potentiostatic electrochemical impedance spectroscopy (PEIS), along with surface-sensitive X-ray photoelectron spectroscopy (XPS) to analyze a 20 nm thick lithium layer over time. The findings reveal that the SEI consists of Li₂S, LiCl, and Li_xP, with a gradient of Li_xP present. The authors suggest that highly reduced species, such as Li₃P, form near the metallic lithium interface, while delithiated Li_xP species develop in contact with LPSCl. A continuous growth of the SEI is attributed to the reaction between LPSCl and Li₃P to form Li_xP, which persists as long as phosphorus percolation pathways remain in the SEI. The study emphasizes the importance of accurately identifying SEI species and understanding their formation and stability.

We would like to thank the reviewer for recognising the novelty and importance of our work. Their feedback reassures us that we were able to clearly convey the intended message.

However, some of the following aspects require further clarification.

(1) Introduction: Please reformulate that lithium non-uniformity in liquid-based batteries can be circumvented by the application of solid electrolytes due to their mechanical strength. This formulation appears misleading to me, as it leaves out that this is only possible with homogeneous stack pressure application. Without it, solid-state batteries also suffer inhomogeneous Li-plating.

Reworded as requested:

“In theory, this issue can be mitigated by applying a homogeneous stack pressure with a solid electrolyte (SE) with sufficient mechanical strength.[4]”

(2) Introduction: Is it really just the “low” ionic conductivity of oxide solid electrolytes, or also their bad interface contact that plays a major role in the challenges related to their commercialization (which relates back to their mechanical properties)? Also this formulation appears misleading to me.

While the ionic conductivity is arguably the ultimate showstopper from a performance perspective, the reviewer is correct in pointing out that oxides face multiple additional challenges, including their brittleness and high density. We have edited the text to reflect the reviewer’s point as follows:

“Several Li-ion conducting SEs have been discovered.[5] Oxides exhibit the lowest theoretical reduction potentials and, therefore, the greatest stability against Li metal anodes.[6] However, their ionic conductivities have so far been too low to be used as the catholyte in practical applications,[5] whilst oxide’s poor solid–solid contacts between the Li metal and a rigid oxide solid electrolyte hinder its use as a separator.[7] In contrast, softer sulphides have demonstrated ionic conductivities exceeding those of liquid electrolytes but suffer from limited electrochemical stability windows and poor compatibility with metallic lithium.[5]”

(3) In Figure 1c, where do you fit your R_{SEI} ? It is not clear to me from which resistance in your equivalent circuit you derive the R_{SEI} values, that you plot in Figure 1d.

To make this clearer we have added a supplementary note 3 to the supporting information:

“The SEI is modelled as a long open uniform distributed transmission line, which can be approximated as a M_a element. The relationship between the impedance from a M_a element and the resistance of the SEI (R_{SEI}) is shown in Equation S10.”

$$Z_{M_a} = R_{SEI} \frac{\coth[(\tau j\omega)^{\alpha/2}]}{(\tau j\omega)^{\alpha/2}}$$

We now refer to this in the main text as follows:

“The M_a element is used to approximate the transmission line element and calculate the ionic resistance of the SEI (R_{SEI}) using Equation S10 shown in Supplementary Note 3.”

We have now also clarified that R_{SEI} comes from within M_a in the caption for Figure 1d. We have also clarified in the caption of Figure S2 that the resistance of the transmission line is the R_{SEI} .

(4) How do the authors translate their observations of continuous SEI growth to the scenario of “zero-lithium-excess” solid-state batteries? In a real setup of “zero-lithium-excess”, we would not have a continuous reaction front between metallic lithium and LPSCI. On the contrary, as we would form species of less reduced phosphorus (P, LixP) first between LPSCI and the current collector (see CV in Figure 5a), and after that plate lithium. Would that impact the stability of the SEI and limit its continuous evolution?

The reviewer is right in interpreting the results. In a “zero-lithium-excess” scenario, first less reduced phosphorus would form next to the current collector, then fully reduced phosphorus. This is what we ascribe to the reduction current for most of the first step of the CTTA experiment, shown in Figure R1. We see no reason why this would impact the stability and continuous evolution of the SEI.

Figure R1. The first coulometric titration step of CTTA in the three electrode (a) and two electrode (b) setups, revealing a significant capacity is used to reduce the electrolyte through different potentials, before the Li potential is reached.

Upon discharge in a “zero-lithium-excess” configuration, once all the Li metal is consumed, the phosphorus in the SEI would become oxidised. At this point the SEI would cease to grow, although not be converted back to the SE. Upon charge the phosphorus would become lithiated again and the SEI would continue to grow.

(5) The low signal-to-noise ratio on the P2p spectra in Figure 3a and 4c makes it difficult to conclude on the different Li_xP species and their accurate binding energy positions. Did you calculate the error that P2p induces in your quantitative analysis in Figure 4d?

The peak fitting uncertainties used to generate the error bars in Fig. 3c were estimated using CasaXPS where random noise is introduced to a synthetic peak model to assess the impact this has on the determined peak areas. This estimate relies on the assumption that the noise in the measured data obeys a Poisson distribution and the standard deviation of the residual when fitting the background is close to unity. The synchrotron SOXPES and HAXPES data unfortunately do not meet these criteria so a quantitative assessment of the uncertainty is not possible.

To improve the signal-to-noise ratio and determine more accurate binding energy positions for Li_xP species we conducted an additional experiment where we sputtered phosphorus on top of $\text{Li}_6\text{PS}_5\text{Cl}$, to create an initially pure phosphorus SEI (Figure 4). We then in situ lithiate this layer and observe the layer at resting conditions post lithiation. We again observe phosphorus dynamically changing oxidation state, becoming progressively more oxidised ($\text{Li}_3\text{P} \rightarrow \text{Li}_x\text{P} \rightarrow \text{P}$). This is conclusive proof that lithiated phosphorus is not stable against $\text{Li}_6\text{PS}_5\text{Cl}$.

To reflect this added experiment, we have added the following to the text:

“Due to the low signal-to-noise ratio of the phosphorus spectra, a second experiment was conducted. Here, phosphorus was directly sputtered onto the $\text{Li}_6\text{PS}_5\text{Cl}$ within the XPS analysis chamber, before being lithiated by VEP to a capacity of $51.7 \mu\text{Ah}$ ($\sim 0.26 \text{ mAh cm}^{-2}$) and finally left at rest for 36.8 h (Figure 4a). Prior to VEP ($0 \mu\text{Ah}$) the S 2p spectrum is dominated by $\text{Li}_6\text{PS}_5\text{Cl}$ at 161.4 eV, with minor peaks attributable to P-S-P bonding and Li_2S , demonstrating that the sputtering process causes negligible electrolyte degradation. This is confirmed by the Cl 2p, S 2p and Li 1s XPS spectra gathered during phosphorus sputtering in Supplementary Fig. 7, which show no shifts in binding energy. A corresponding $\text{Li}_6\text{PS}_5\text{Cl}$ peak is identified in the $0 \mu\text{Ah}$ P 2p spectrum at 131.8 eV and the major peak at 129.7 eV is attributed to P^0 . The minor peak at 133.2 eV is assigned to species containing P-O bonding. During lithiation the S 2p Li_2S peak grows in relative intensity while the $\text{Li}_6\text{PS}_5\text{Cl}$ peak reduces in intensity, indicative of electrolyte reduction. The proportion of Li_2S increases rapidly from 4 at.% S

prior to lithiation to 65 at.% S after 6.7 μAh , reaching 87 at.% S after 51.7 μAh (Fig. 4b). At the same time the point of maximum intensity in the P 2p region shift to lower binding energy and peak fitting after 51.7 μAh reveals three additional species at 127.7, 126.9 and 125.8 eV, attributed to Li_xP , Li_yP and Li_3P , respectively, where $0 < x < y < 3$. The evolution of these phosphorus-containing species in Fig. 4c reveals that lithiation progresses through $\text{P}^0 \rightarrow \text{Li}_x\text{P} \rightarrow \text{Li}_y\text{P} \rightarrow \text{Li}_3\text{P}$ and the proportions of the most lithiated phases increase with increasing capacity. Interestingly, significant electrolyte reduction was observed in the S 2p spectra before Li_yP and Li_3P were detected, suggesting that even the least lithiated Li_xP species can reduce $\text{Li}_6\text{PS}_5\text{Cl}$.

Figure R2. Phosphorus lithiation. a) Evolution of the S 2p and P 2p spectra during VEP-XPS lithiation of a sputtered phosphorus layer on top of a $\text{Li}_6\text{PS}_5\text{Cl}$ pellet using an ebeam current of 30 μA to a capacity of 51.7 μAh ($\sim 0.26 \text{ mAh cm}^{-2}$), followed by resting under OCV for 36.8 h. All spectra have been normalised for clarity. b) Breakdown of sulphur-containing species identified in a). c) Breakdown of phosphorus-containing species identified in a). Fitting details are provided in Supplementary Table 8.

After lithiation the sample was left to rest at OCV within the XPS analysis chamber. Fig. 4a and b reveal negligible changes in the sulphur-containing species during this period, suggesting Li_2S is stable. On the other hand, during rest the maximum intensity in the P 2p spectra shifts back to higher binding energy and the breakdown of phosphorus-containing species in Fig. 4c reveals that spontaneous delithiation occurs according to $\text{Li}_3\text{P} \rightarrow \text{Li}_y\text{P} \rightarrow \text{Li}_x\text{P} \rightarrow \text{P}^0$. The final oxidation of Li_xP to P^0 must be accompanied by an additional reduction reaction, suggesting this delithiation process will lead to further electrolyte decomposition.

These observed trends in the P 2p spectra are mirrored in the Li 1s spectra in Supplementary Fig. 8, while no changes are observed in the Cl 2p spectra. In the O 1s spectra it is evident that a Li_2O peak at 528.4 eV emerges during lithiation and a peak at 531 eV grows during rest, indicative of the accumulation surface oxygen species. While the P-O peak in Fig. 4a also grows in intensity during rest

it still remains only a minor phosphorus-containing species (Fig. 4c), so side reactions with gases inside the XPS chamber are not expected to have a significant impact on the observed phase evolution.”

The higher-resolution data from this experiment has also enabled us to refine the peak assignments in Figure 3. We now observe that Li_3P is present only initially, before transforming into Li_yP , which then gradually evolves into Li_xP .

(6) Please list your XPS fitting parameters for S2p, P2p and P2s in the supporting information (full width at half maximum and binding energy positions of the fitted compounds).

All fitting parameters have now been included in the Supplementary Information as Supplementary Tables 7–10.

(7) How do you explain your Li1s spectra after 22h in Figure S11a? The component that you previously assigned to Li_2O is shifted and split in two components now – could that indicate other species forming after such a long time instead of Li being only consumed by SEI growth?

In Supplementary Fig. 11a it is not possible to accurately assign a single Li 1s peak for every lithium-containing species, so all species containing lithium in the +1 oxidation state have been lumped together. Immediately after lithium evaporation only a single broad Li^+ peak can be identified, which will include Li_2S , LiCl , Li_xP , Li_3P , $\text{Li}_6\text{PS}_5\text{Cl}$, LiOH and Li_2O . After 22 h the additional peak at approximately 55 eV is likely indicative of the formation of Li_2CO_3 , which is known to accumulate on the surface of both $\text{Li}_6\text{PS}_5\text{Cl}$ and lithium metal.

We agree with the reviewer that lithium will be consumed from both sides as it will react with both the solid electrolyte and residual gases inside the XPS chamber.

(8) Your assignment of the peaks in the CV of Figure 5a is not clear to me. If I understood correctly, you assign peak "a" to the formation of Li_xP , and peak "b" to their lithiation until Li_3P . I think peak "a" should be the formation of Li_2S and partially P, and peak "b" should be Li_xP to Li_3P formation, which is reversed on peak "c". As there is no further peak upon oxidation, and in subsequent reduction peak "a" has disappeared, it would make sense to correlate peak "a" to the formation of Li_2S , which is not reversed at such low potentials.

There appears to be a little bit of confusion here. We do assign peak **a** to the formation of Li_2S , but also some partially lithiated P (see Equation 1) and we do assign peaks **b** and **c** as the reviewer describes (see Equations 2 and 3 respectively). Please see the following section of the manuscript:

“During the first reductive sweep, two clear peaks can be observed (labelled a and b in Figure 5a). Peak a can be ascribed to the reduction of argyrodite. The onset potential of 1.13 V vs. Li^+/Li is below the computationally predicted value of 1.71 V vs. Li^+/Li , [6] yet significantly higher than previously thought. [23] As reduction initially occurs above the theoretical full reduction potential of phosphorus to Li_3P (0.87 V vs. Li^+/Li), this peak can be assigned to the reduction of $\text{Li}_6\text{PS}_5\text{Cl}$ to form Li_2S , LiCl and partially lithiated phosphorous species which can be summarised as Li_xP , where $x < 3$ (Equation 1). Peak b can be attributed to the lithiation of Li_xP to form the fully reduced Li_3P (Equation 2). Indeed fitting the areas of peaks a and b yields a capacity ratio of $\sim 3:1$ (Figure S13), which would be consistent with a conversion of LiP to Li_3P (Equations 1 and 2). The total capacity of the reductive sweep is 11.1 mC cm^{-2} , which would correspond to an SEI thickness of 28 nm (if fully dense and consisting of only Li_2S , LiCl and Li_3P).

During the oxidative sweep, a peak can be seen (labeled c) at 0.87 V vs. Li⁺/Li, in line with the reoxidation of Li₃P to Li_xP (Equation 3).^[6, 20, 24] Interestingly this occurs at a potential that is below the observed initial Li₆PS₅Cl reduction potential. On subsequent cycles (Figure 5b) peak a is no longer clearly visible suggesting that most of the SEI growth occurs on the first reductive sweep. Nonetheless, an increase in the peak current of peak c with cycling is indicative of a continuous growth of the SEI.”

(9) What about the reversibility of such SEI layer? From CV, I can tell that the phosphorous species form reversibly. Did the authors study with CTTA, EIS and XPS (or other techniques) also the reversibility of such SEI growth or could elaborate on the implications on the SEI composition and stability upon subsequent battery cycles?

A potential cut-off control would be necessary to ensure that a sufficient amount of lithium remains in the system. Without this safeguard, the potential of the negative electrode would gradually drift upward, even under open-circuit voltage (OCV) conditions, making it difficult to determine the state of charge (SoC) of the cathode active material. This uncertainty could ultimately compromise cell performance and lead to failure. Consequently, in a practical battery system, cycling of the solid electrolyte interphase (SEI) would not be expected to occur.

(10) Minor side note on typos: Please always leave the reference [x] either before or after your punctuation, it appears to be mixed sometimes. Figure S5, please indicate which is your data and what is the reference in the labels for clarity.

We have addressed the referencing inconsistency and revised the caption of Figure S5 to improve clarity, in accordance with the reviewer’s request.

Reviewer #2

This study aims to understand the role of phosphorus in the diffusion-controlled growth of the interphase using electrochemical and X-ray photoelectron spectroscopy techniques. While this topic is of significant interest in sulfide-based all-solid-state batteries, neither the findings nor the employed techniques appear to be novel. The methodology closely resembles that of a study reported by the Janek group (Nat Commun 14, 6946 (2023)), and the findings have already been presented by the same research group in a previous work (Nat Commun 13, 7237 (2022)). Additionally, similar conclusions were drawn in ACS Energy Lett. 2024, 9, 7, 3492–3500, where hard X-ray photoelectron spectroscopy (HAXPES) was used. Furthermore, although not specifically focused on Li₆PS₅Cl, the evolution of the solid electrolyte interphase and its chemical composition has been investigated using operando X-ray photoelectron spectroscopy in Nat Commun 9, 2490 (2018). Given these considerations, I doubt whether this study meets the standards expected for publication in Nature Communications.

This reviewer references four papers to suggest there is a lack of novelty for the standards expected for *Nature Communications*. We will tackle them here one by one.

1. Nat Commun 14, 6946 (2023). Aktekin et al. comes up with the idea of CTTA and concludes that the SEI continually grows via an undiagnosed diffusion mechanism.

We expanded the CTTA to include an impedance step to further study the SEI properties. Our manuscript generally agrees with the above manuscript, but crucially we provide the previously undiagnosed diffusion mechanism (diffusion through phosphorus). We highlight the formation of a gradient of lithiated phosphorus in the SEI, which highlights that the rate of growth of the SEI is not

completely linear with the root of time in $\text{Li}_6\text{PS}_5\text{Cl}$ as suggested by Aktekin *et al.*. Additionally, we suggest a route to stop the continuous SEI growth showing some very promising preliminary results.

2. Nat Commun 13, 7237 (2022) – Narayanan et al. (our group) investigate the effect of current density of SEI formation at the lithium $\text{Li}_6\text{PS}_5\text{Cl}$ interface, suggesting a higher current density leads to the formation of Li_3P at the Li metal interface, whereas a lower current density leads to the formation of Li_xP at the Li metal interface.

Whilst Narayanan et al. observe the formation of Li_3P and Li_xP in the SEI, the reasoning for their formations is not the same. If current density was low, Narayanan et al. initially observed Li_xP formed at the current collector interface, whilst if it was high they observed Li_3P . With the conclusion being to use a high current density to prevent Li_xP formation.

Our manuscript reveals that the reason for the lack of Li_xP observation at high current density, is due to there not having been enough time for the Li_3P formed at high current density to transform into Li_xP . Our work shows that Li_3P will always be present on the Li Metal anode, whilst Li_xP will also be present on the SE side of the SEI, with a gradient of lithiation existing between the two. No variation in current density will change this. This is fundamentally a different conclusion to Narayanan *et al.*

3. ACS Energy Lett. 2024, 9, 7, 3492–3500 - Aktekin et al. uses HAXPES operando to study SEI formation with a 6 nm Ni working electrode. They observe reduction of $\text{Li}_6\text{PS}_5\text{Cl}$ at 1.75 V vs. Li^+/Li . They report a layered SEI, with Li_2S near the current collector and phosphorus and chlorine compounds buried underneath.

With the exception of the agreement of a reduction potential of $\text{Li}_6\text{PS}_5\text{Cl}$ of circa 1.7 V vs. Li^+/Li , the results of our manuscript are different. Whilst Aktekin et al. are unable to probe the phosphorus environment in their setup, to which they suggest a layered SEI structure with phosphorus buried under Li_2S , our manuscript reveals this not to be the case. We see phosphorus throughout the thickness of the SEI and observe different oxidation states traversing the thickness. We observe this oxidation state also changing through time at OCV, revealing that phosphorus plays a dynamic role in the SEI growth. This is in complete contrast to Aktekin et al., who fail to observe phosphorus after the initial $\text{Li}_6\text{PS}_5\text{Cl}$ reduction.

We understand that our phosphorus signal may be weak, which is why we have conducted an additional experiment (see our response to Reviewer 1 point 5). We sputtered pure phosphorus on top of $\text{Li}_6\text{PS}_5\text{Cl}$, to create an initially pure phosphorus SEI. We then in situ lithiate this layer and observe the layer at resting conditions post lithiation. We again observe phosphorus dynamically changing oxidation state, becoming progressively more oxidised ($\text{Li}_3\text{P} \rightarrow \text{Li}_x\text{P} \rightarrow \text{P}$). This is conclusive proof that lithiated phosphorus is not stable against $\text{Li}_6\text{PS}_5\text{Cl}$.

4. Nat Commun 9, 2490 (2018) - While this paper does look at the SEI of a phosphorus containing SE using operando VEP-XPS, their findings are substantially different. Firstly they conclude there is a layered SEI, with Li_2S on the SE side and lithiated phosphorus nearer the current collector (note this is the exact opposite conclusion to ACS Energy Lett. 2024, 9, 7, 3492–3500), they conclude that while Li_3P can be redox active it is stable at rest in the SEI and they do not observe a continual growth of the SEI. This is all in addition to the SE being $\text{Li}_2\text{S}-\text{P}_2\text{S}$, rather than $\text{Li}_6\text{PS}_5\text{Cl}$ and there being a significant oxygen proportion to their SE with causes side reactions to occur.

We have now added the following paragraph to the introduction and the discussion, to make our differences clearer.

“Work by the Janek group challenged this assumption, reporting continuous SEI growth, which they described using the Wagner model for diffusion-controlled solid-state reactions.[10-12] Yet the underlying diffusion mechanism was not understood.”

“However, previous HAXPES research has failed to see this gradient due to an inability to observe phosphorus in the SEI due to a Ni current collector and lithium oxide attenuating the beam,[26] while previous VEP-XPS measurements have failed to observe the instability of lithiated phosphorus with the solid electrolyte due to either a limited observation time or different SE being studied.[14, 27]”

In our work we use HAXPES and SOXPES to show that lithiated phosphorus transverses the SEI, being more lithiated nearer the current collector and less lithiated nearer the SE. We show that Li_3P and Li_xP are not stable at rest in the SEI and react with the underlying SE, causing the continual growth of the SEI. The key insight that phosphorus drives the continual growth of the SEI has never been shown or concluded before. This novel finding allows routes to prevent the continual SEI growth to be found and we show some very promising preliminary results on this in this manuscript.

Reviewer #3

In this manuscript, the authors aim to elucidate the growth mechanism of the solid electrolyte interphase (SEI) formed at the interface between the sulfide-based solid electrolyte $\text{Li}_6\text{PS}_5\text{Cl}$ and lithium metal. To achieve this, they employed a combination of techniques including electrochemical measurements, X-ray photoelectron spectroscopy (XPS), and both hard and soft X-ray photoelectron spectroscopy (HAXPES and SOXPES). Particular focus was placed on the presence of lithium phosphide species (Li_xP) within the SEI and their potential redox behavior during SEI formation.

The experimental approach is systematic and the methodologies used are appropriate. However, the findings largely overlap with previously reported results and do not offer substantial new insights.

Please see our opening statement and the more detailed comments to Reviewer 2.

More critically, the photoelectron spectroscopy analysis—which constitutes the central component of this work—contains fundamental errors in both its interpretation and application of the technique. These issues require careful reevaluation from the level of basic principles.

For example, the core-level P 2p spectra were fitted using a single peak, despite the well-established fact that 2p orbitals must be analyzed as a spin-orbit doublet. There are no exceptions to this requirement.

In light of these issues, I regret to conclude that this manuscript is not suitable for publication in its current form.

All P 2p spectra were, of course, fitted using doublets, applying the standard area ratio of 2:1 ($2p_{3/2} : 2p_{1/2}$) and a spin-orbit splitting of 0.87 eV. For clarity, we presented only the sum of the fitted doublets in the figures. We have now included the individual doublets in Figure 3a as dotted lines to avoid confusion and have updated the figure caption accordingly.

Figure 3. Virtual-electrode XPS. a) XPS of phosphorous in the SEI after $\sim 0.01 \text{ mAh cm}^{-2}$ of lithium was VEP through a $\text{Li}_6\text{PS}_5\text{Cl}$ pellet over 1 hour using an ebeam current of $2.5 \mu\text{A}$ ($\sim 0.01 \text{ mA cm}^{-2}$). All P 2p spectra are fitted as doublets (dotted lines), with shaded areas being summed for clarity and fitting details are provided in Supplementary Table 7.

All fitting parameters have now been included in the Supplementary Information as Supplementary Tables 7–10.

Given the severity of the reviewer’s statement, we respectfully request that they provide specific details regarding the “careful reevaluation” they are suggesting.

Response to Reviewers' Comments

We would like to thank the reviewers for their insightful comments and appreciation for our work. Changes implemented to the original document are highlighted in yellow for the direct quote to the revised manuscript.

Reviewer #1

First, I wish to thank the authors for addressing my comments and questions accordingly and in great detail. They have enhanced the clarity and significance of the manuscript, especially with additional experiments on the lithiated phosphorus evolution using VEP-XPS.

I have three comments concerning the author's changes and answers to my previous review:

(1) For clarity, please bring component P-S-P in Figure 4a to the front, or the color-code is not clear anymore.

Done as requested.

(2) Please stay consistent with the labelling for the lithiated phosphorus species. In Figure 4 they are split in Li_xP, Li_yP and Li₃P, but in Figure 5 they are summarized as Li_xP and Li₃P, though the fitting indicates three species.

Done as requested.

(3) Thank you for the clarification on Figure 6. Indeed, it is confusing to read the manuscript description and to look at different species highlighted in Figure 6, which come from theoretical predictions. If the authors agree, I would suggest unifying the species assignment displayed in Figure 6 with the description provided below the graph.

Whilst we appreciate the suggestion, we are concerned that implementing it may introduce further confusion, given the ambiguity of presenting only Li_xP , an issue already highlighted in the reviewer's earlier comment. We have therefore respectfully decided not to adopt this suggestion. Our intention is not to imply that all the phosphorus species shown in Figure 6a are experimentally observed, but rather to indicate the potentials at which they are thermodynamically favoured. This has also been clarified in the figure legend, where we explicitly state that the displayed potentials represent theoretical stability windows.

Reviewer #2

The authors have made a commendable effort to address previously reported studies, including those, I highlighted as similar, and have provided convincing arguments to underscore the novelty of their work. However, there remain several points that require further clarification and deeper investigation to strengthen the overall impact and scientific rigor of the study:

1. The current study focuses on an argyrodite-type sulfide solid electrolyte. Would the same interphase growth mechanism and SEI evolution behavior be expected for other classes of sulfide-based electrolytes? In particular, I am curious about how the role of phosphorus in the SEI might vary depending on the structural nature of the anionic units, such as ortho-

thiophosphates (PS43-), pyro-thiophosphates (P2S74-), hypo-thiophosphates (P2S64-), and meta-thiophosphates (P2S62-). While I understand the scope of this work is centered on a specific electrolyte type, a brief discussion of how the findings might generalize to other chemistries would enhance the broader relevance of the study.

The reviewer raises a valid point. Wenzel, Janek et al. (Solid State Ion., 286, 24–33 (2016)) have seen a similar behaviour with $\text{Li}_7\text{P}_3\text{S}_{11}$ (which contains both ortho and pyro-thiophosphates). They observe the loss of Li_3P in the SEI whilst waiting after the deposition of Li, however, they only speculate that it may have reacted with trace amounts of oxygen or water in the UHV XPS chamber. The data they present however, indicates to us that the phosphorus plays a similar role in the SEI formed between Li and $\text{Li}_7\text{P}_3\text{S}_{11}$ as it does in the SEI formed between Li and $\text{Li}_6\text{PS}_5\text{Cl}$. In addition, yet unpublished work from the El Kazzi's group (10.26434/chemrxiv-2025-spbvc) shows the existence of multiple degrees of lithiation of phosphorus in the SEI of Li_3PS_4 (another ortho-thiophosphate) indicating to us that for this electrolyte phosphorus plays the same role in the SEI.

To highlight these points, we have added the following lines to the manuscript:

“Although this work has only looked at the solid electrolyte $\text{Li}_6\text{PS}_5\text{Cl}$, we believe this continuous mechanism of SEI formation would be present in any solid electrolyte that can be reduced to contain partially lithiated phosphorus, or potentially any element that can alloy with lithium. Indeed, a loss of Li_3P in the SEI at rest has been observed over time with $\text{Li}_7\text{P}_3\text{S}_{11}$.^{\cite{Wenzel2016}} While this observation was speculated to be due to reactions with trace amounts of oxygen or water in the UHV XPS chamber, our results indicate that it may be exhibiting SEI evolution similar to that of the SEI of $\text{Li}_6\text{PS}_5\text{Cl}$. More recently El Kazzi's group has shown multiple degrees of lithiation of phosphorus in the SEI of Li_3PS_4 ,^{\cite{Siller2025}} indicating that phosphorus may play the same role in its SEI.”

Plus the following line to the abstract:

“We believe that this growth mechanism would apply to any SEI that can contain partially lithiated phosphorus, or potentially any lithium alloy.”

2. The use of distribution of relaxation times (DRT) analysis would significantly enhance the interpretation of the EIS Nyquist plots. I strongly encourage the authors to perform DRT analysis and relate the resulting features to the evolution of the SEI. This would provide deeper insight into the individual EIS plots contributing to the interfacial processes.

Whilst we agree that DRT can be a valuable tool, it relies on the assumption that the impedance data can be represented solely by a series of R/C circuits. In our analysis, we deliberately developed an equivalent circuit model tailored to the physical characteristics of the system. Specifically, we employed a transmission line model, which more accurately reflects the multicomponent and spatially distributed nature of the SEI. As a result, our fitting model does not conform to a purely R/C-based structure. Nevertheless, we will provide the raw impedance data alongside the manuscript, allowing future readers to perform DRT analysis should they wish.

3. The manuscript suggests that lithium diffusion through a percolated Li_xP phase is the rate-limiting step. However, this assumption relies on an unverified comparison of the diffusivities of Li_xP and Li_3P . Could the observed non-linear growth behavior instead (or additionally) stem from evolving SEI porosity, phase morphology, or changes in percolation connectivity? A more

quantitative or at least qualitative discussion on this aspect would strengthen the proposed mechanism.

The diffusivity values we report are taken from a previously published GITT study on lithium in Li_xP and Li_3P conducted by our group (10.1016/j.matt.2020.09.017). The reviewer is correct in noting that an evolving SEI could influence component volume fractions and porosity, potentially altering the percolation pathway. We have previously addressed this aspect in a recent publication from our group, where we compared SEI growth behaviour in LAGP and LGPS (10.1039/D4CC04462B), and we have also commented on this point in the following paragraph of the manuscript:

“Two important points should be noted. First, this process relies on the presence of a phosphorus percolation pathway throughout the SEI, highlighting the critical role of SEI nanostructure. Indeed, changing the chemical composition of argyrodite without significantly changing the volume fraction of phosphorus in the SEI (Table S1-4), showed minimal changes to the kinetics of SEI growth. \cite{Guo2024} While the ability to lithiate phosphorous in the SEI was found to be key to the difference in growth kinetics between $\text{Li}_{10}\text{GeP}_2\text{S}_{12}$ (LGPS) and $\text{Li}_{1.5}\text{Al}_{0.5}\text{Ge}_{1.5}(\text{PO}_4)_3$ (LAGP). \cite{Liang2024} Meanwhile, the graded structure of the SEI, which ensures that the most reduced phosphorous species are not in physical contact with the solid electrolyte, has been linked to the superior stability of the LiPON SEI. \cite{Turrell2025} Second, if a continuous percolation pathway is present, SEI growth will persist as long as metallic lithium is available.”

To further clarify this point, we have included the following line in the manuscript:

“This variation in diffusivity could be due to the variation of diffusivity of Li in P with lithiation state, [20] the change of the phosphorus volume fraction in the SEI through lithiation states (Table S1-2), or an evolving porosity.”

4. The reversible delithiation observed via VEP-XPS and voltammetry is a central part of the proposed instability mechanism. However, its implications for long-term cycling behavior are not fully addressed. How reversible is this delithiation beyond the immediate OCV rest period? Does this process result in partial dissolution or dynamic reformation of the SEI over extended cycling? Clarifying this would be important for understanding the stability and durability of the interphase.

The key implication of our study is that, as long as lithium metal is present, the SEI will continue to grow due to phosphorus, leading to increased impedance and self-discharge. This process occurs effectively at any state of partial charge. Regarding the reversibility of SEI delithiation, our cyclic voltammetry data demonstrate that the delithiation of phosphorus is indeed reversible: peak c in Figure 6b grows progressively with each cycle, indicating a continuously expanding SEI in which more lithiated phosphorus is oxidized during each subsequent cycle. There is no reason to expect that this behaviour would differ at open-circuit. From an application standpoint, however, this reversibility is of limited significance, as both impedance growth and self-discharge persist regardless.

5. All experiments in the manuscript appear to have been conducted under a stack pressure of 13 MPa. Have the authors considered conducting similar tests under lower or variable pressures to assess the sensitivity of SEI evolution to mechanical loading? Such data would greatly enhance the relevance of this work for practical battery integration, where high stack pressures may not always be feasible.

Whilst, in theory, it may seem reasonable to conduct our electrochemical experiments (such as CTTA) under different pressures, in practice the primary effect of varying pressure is to alter the electrochemically active area, as recently demonstrated by the Kovalenko group (10.1038/s42004-025-01496-0). This change would obscure any potential effects of mechanical loading on the SEI, making it difficult to isolate the influence of pressure on SEI properties.

It should be noted that whilst our CV, CTTA and related experiments are conducted at 13 MPa, our VEP-XPS and SOXPES/HAXPES are all conducted with no applied pressure in a vacuum. The fact that our electrochemical and XPS results indicate the same SEI evolution, leads us to conclude that mechanical loading does not significantly change the sensitivity of SEI evolution.

Reviewer #3

This manuscript is worthy of publication once the remaining revisions requested by the editorial office have been addressed.

We thank the reviewer for their time and effort in reviewing our manuscript.